# Performance of cardiopulmonary exercise testing for the prediction of post-operative complications in non cardiopulmonary surgery: A systematic review

**Daniel J. Stubbs, Lisa A. Grimes, Ari Ercole** *

University Division of Anaesthesia, Department of Medicine, Addenbrooke's Hospital, Hills Road, Cambridge, CB2 0QQ, Cambridge, United Kingdom

* ae105@cam.ac.uk

## Abstract

### Introduction

Cardiopulmonary exercise testing (CPET) is widely used within the United Kingdom for pre-operative risk stratification. Despite this, CPET's performance in predicting adverse events has not been systematically evaluated within the framework of classifier performance.

### Methods

After prospective registration on PROSPERO (CRD42018095508) we systematically identified studies where CPET was used to aid in the prognostication of mortality, cardiorespiratory complications, and unplanned intensive care unit (ICU) admission in individuals undergoing non-cardiopulmonary surgery. For all included studies we extracted or calculated measures of predictive performance whilst identifying and critiquing predictive models encompassing CPET derived variables.

### Results

We identified 36 studies for qualitative review, from 27 of which measures of classifier performance could be calculated. We found studies to be highly heterogeneous in methodology and quality with high potential for bias and confounding. We found seven studies that presented risk prediction models for outcomes of interest. Of these, only four studies outlined a clear process of model development; assessment of discrimination and calibration were performed in only two and only one study undertook internal validation. No scores were externally validated. Systematically identified and calculated measures of test performance for CPET demonstrated mixed performance. Data was most complete for anaerobic threshold (AT) based predictions: calculated sensitivities ranged from 20-100% when used for predicting risk of mortality with high negative predictive values (96-100%). In contrast, positive predictive value (PPV) was poor (2.9-42.1%). PPV appeared to be generally higher for cardiorespiratory complications, with similar sensitivities. Similar patterns were seen for the

**Data Availability Statement:** All relevant data are within the manuscript and its Supporting Information files.

**Funding:** DJS is supported by a Wellcome Trust Clinician PhD Fellowship (204017/Z/16/Z - https://wellcome.ac.uk/funding/schemes/clinical-phd-programmes). The funders had no role in study design, data collection and analysis, decision to publish, or preparation of the manuscript.

**Competing interests:** The authors have declared that no competing interests exist.

association of Peak $VO_2$ (sensitivity 85.7-100%, PPV 2.7-5.9%) and $VE/VCO_2$ (Sensitivity 27.8%-100%, PPV 3.4-7.1%) with mortality.

## Conclusions

In general CPET's 'rule-out' capability appears better than its ability to 'rule-in' complications. Poor PPV may reflect the frequency of complications in studied populations. Our calculated estimates of classifier performance suggest the need for a balanced interpretation of the pros and cons of CPET guided pre-operative risk stratification.

## Introduction

There is increasing focus on the identification of the 'high-risk' surgical patient. Large scale national projects such as the 'National Emergency Laparotomy Audit' (NELA) incorporate this pre-operative assessment of risk into pathways to stratify perioperative care [1]. In the elective setting techniques including risk scales, patient function, organ biomarkers, and levels of comorbidity have all been used with the aim of risk stratifying patients [2]. Risk stratification may facilitate precision interventions to improve outcome in this patient cohort, as well as assist in the process of shared decision making. Cardiopulmonary exercise testing (CPET) is attractive as a basis for patient stratification as it represents an objective and holistic physiological assessment of cardiopulmonary reserve. Briefly, an individual exercises (typically on a cycle) in a graded manner whilst simultaneous measurements of inspired, and expired gases, heart rate, and electrocardiogram are made [3]. Numerous objective summaries of cardiopulmonary function can be derived from CPET including the oxygen consumption at the anaerobic threshold (AT), maximal oxygen consumption ($VO_2Max$, a plateau in oxygen uptake at maximal exercise levels), 'Peak $VO_2$',(or the highest oxygen uptake recorded in a sub-maximal test), as well as ventilatory equivalents for both oxygen ($VE/VO_2$) and carbon dioxide ($VE/VCO_2$) [4].

In 1993 it was reported that an AT of <11ml/kg/min accurately identified patients at risk of cardiovascular mortality undergoing intra-abdominal surgery [5]. Subsequent work by the same group utilised this finding to guide perioperative care, with high risk patients undergoing major surgery allocated to high dependency (HDU) or intensive care unit (ICU) beds [6]. Importantly, this work also demonstrated that a single metric alone may not be adequately discriminating. Patients with AT of >11mL/Kg/min but with evidence of myocardial ischaemia during CPET testing, or with severe pulmonary disease also suffered cardiopulmonary mortality and the overall numbers of deaths was low. Patients deemed to be 'fit' on CPET with an AT of >11mL/Kg/min and with no myocardial ischaemia had no instances of cardiac mortality. The authors themselves suggest that CPET is "even more reliable at detecting those not at risk..." [6].

More recently, CPET has been used to characterise individuals with chronic cardiac or respiratory failure [7] and, especially in the United Kingdom (UK), is being increasingly used for the stratification of patients prior to high risk procedures [8]. With nearly 15,000 individual tests a year from 2011 [9], this represents a significant logistical, organisational, and financial undertaking with the National Institute for Health and Care Excellence (NICE) quoting an average unit cost of £183 [10].

CPET is often used to gate access to more advanced care (such as intensive care unit admission) [6]. This strategy, of referring those identified as high risk using CPET to elective critical

care stay, could have serious consequences if, in fact, the positive predictive value of CPET is poor. Critical care bed capacity is under incredible pressure [11] and unnecessary utilisation could add further strain to an already stretched system. Furthermore lack of critical care beds is already a significant risk factor for day of surgery cancellation [12], if this admission is based on the result of information with poor predictive performance this could lead to an unnecessary delay in an individual's care. The issue of predictive performance is also important when decisions not to operate. Treatment options could be limited based on an individual being judged high risk but this result may in fact poorly predict the occurrence of complications. In a survey of practice the majority of respondents did not recommend cancellations based on CPET results but a significant minority (33%) did or, based on free-text comments, used these tests to inform subsequent discussions between clinical staff and the patient [9]. In their 2016 update on preoperative investigation NICE failed to make specific recommendations regarding the use of CPET but noted that other simpler, and cheaper tests may also be used for risk prediction [10]. For these reasons, the performance of CPET as a prognostic test should be carefully evaluated, as with other tests, in terms of its ability to correctly (or incorrectly) identify patients with and without the outcome of interest [13]. If CPET is used to identify an adverse outcome of interest within some time frame then the statistical interpretation of test performance can be performed using methods familiar from diagnostic testing [14]. Many of these (such as sensitivity, or specificity) are well known. However, no single metric can fully encapsulate a tests performance [15] and it is vital to recognise the potential consequences of false positives and negatives within the specific clinical context.

Numerous systematic reviews of the perioperative utility of CPET have been performed [16–18]. These have highlighted the breadth of current work and concluded that CPET appeared to have promise as a marker of outcome amongst certain specialties [16, 17] but not all [18]. Formal meta-analysis has not been performed due to the heterogeneity of summary data and outcomes, making proper comparison difficult. The use of prediction models in medicine for diagnostic, and prognostic purposes to aid in patient decision making, and targeting of resources to high risk individuals is now widespread [19]. Despite CPET's potential utility in these roles we are unaware of a systematic appraisal of generated prediction models that incorporate information from CPET testing.

In this work we aimed to identify newly published studies assessing CPET's prognostic capabilities for cardiorespiratory complications, 30 day or in-hospital death, and unplanned intensive care unit admission for non-cardiopulmonary surgery. These outcomes were chosen *a priori* based on their relevance to the underlying physiology under test (Cardiorespiratory complications), their importance and examination in initial CPET studies (Mortality) [5], and that elective critical care admission is often advocated for CPET determined high risk cases [6] (unplanned ICU admission). As an extension over previously published systematic reviews we set out to generate comparable metrics of CPET test performance from the published data to aid clinicians in interpreting its performance as a prognostic test. Finally, for studies presenting a risk model using regression techniques, we aimed to provide the first evaluation of the methodological quality of these models using a validated approach [20].

## Materials and methods

### Search strategy

A systematic search strategy was developed in consultation with a medical librarian. We combined keywords for cardiorespiratory complications featured in the European Perioperative Clinical Outcome (EPCO) definitions [21] by the European Society of Anesthesiology (ESA) and the European Society of Intensive Care Medicine (ESICM), death and unplanned ICU

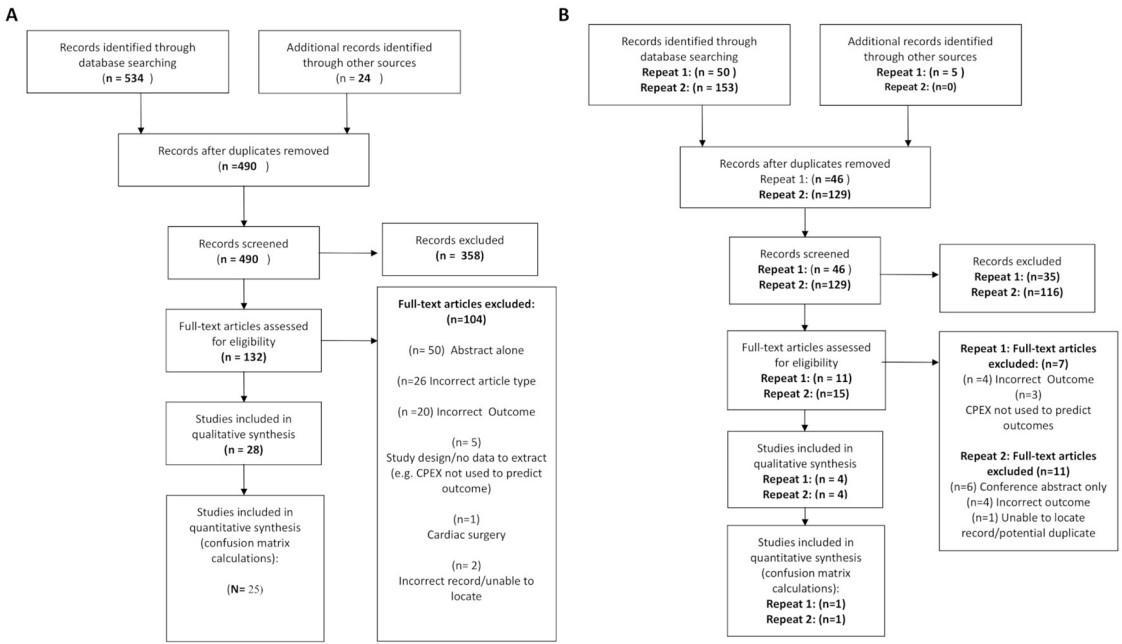

**Fig 1. PRISMA diagram demonstrating identification of included studies.** Panel A demonstrates those included from initial search, Panel B those from repeat search. Meta-analysis refers to those included in calculation of confusion matrix metrics.

admission. Further keywords to identify the perioperative period and cardiopulmonary exercise testing were also used. Medical Subject Headings (MeSH) terms were used as appropriate. Both Medline and Embase databases were searched on March 22nd 2017 using the Ovid interface. Due to the length of time required for study identification and data extraction the search was rerun on July 24th 2018 with new, applicable studies included. After initial submission the search was re-run on 2nd December 2019 for calendar years 2018 and 2019 to ensure our findings were current. PROSPERO registration was obtained (CRD42018095508). Full details of the search strategy is given in S1 Appendix. Titles and abstracts were screened for full-text inclusion by DJS and LAG. In cases of dispute inclusion was determined by arbitration by AE. References of included studies were then searched to identify further studies. Full numbers of studies identified and analysed at each stage are shown in the PRISMA diagram in Fig 1.

## Outcomes

We included studies where CPET was used to identify the risk of the following outcomes. Mortality (in-hospital/peri-operative and within 30 days of surgery), unplanned ICU admission, and complications affecting the cardiorespiratory systems. Studies looking at cardiorespiratory complications must have looked at the occurrence of at least one of the complications featured in the EPCO list of perioperative outcomes [21]. Our initial plan was that each study must follow the strict definitions incorporated within this document. However, it became apparent that this would lead to us including very few studies. Therefore studies were included as long as they looked at the occurrence of a complication featured in the list of cardiorespiratory complications or used one of the cited composite outcome measures (e.g. the Postoperative Morbidity Score—POMS). We included studies where cardiorespiratory complications were presented as a pooled outcome but not if they were included with other organ system complications (e.g. cardiorespiratory complications as judged using POMS was accepted, 'any POMS positive' was not).

## Inclusion and exclusion criteria

Our pre-specified inclusion criteria for analysis were: Any primary research paper using CPET derived variables to aid in identification of adult (aged 18 or over) patients at risk of one of our pre-defined outcomes of interest. Studies could be prospective or retrospective in their design. Patients must have been undergoing non-cardiopulmonary surgery. Our exclusion criteria were: papers where complications were analysed as a pooled outcome, those that were not published in a peer-reviewed journal or available only as an abstract, review articles or correspondence. Only studies written in English were included. Previous meta-analyses on the subject were identified and their references, along with those of included studies, were searched for suitable studies for inclusion.

## Data extraction

Risk of bias was assessed using the 'Quality in Prognostic Studies' (QUIPS) tool [22]. This was performed independently by DJS and LAG with arbitration of any conflicts undertaken by discussion between all three authors. Papers where a multivariable logistic regression model containing CPET derived variables was generated for one of our declared outcomes were also critiqued using the validated prognostic model checklist 'CHARMS' [20]. For all studies, we recorded whether a significant association was found for CPET variables and chosen outcomes, we also noted the statistical methodology used. We also extracted reported sensitivities and specificities for CPET variables. To supplement these published values we also data mined included papers to generate our own confusion matrices. In this process a 'true positive' was defined as a patient who had a CPET value categorising them as high-risk and suffered the outcome in question. A 'true negative' was a patient whose CPET variables categorised them as low-risk and did not suffer the specified adverse outcome. Our generated confusion matrices are available in supplementary material (S1 Table. We also recorded any AUC (area under the receiver operator curve) values for CPET and other prognostic variables reported in the paper. As a secondary aim we undertook the above process for other examined prognostic variables (e.g. American Society of Anesthesiologists- ASA score) within each paper. All data extraction was performed independently by DJS and LAG.

## Prediction performance metrics

After generating confusion matrices for each study, classifier performance was evaluated on the basis of sensitivity, specificity, positive predictive value (PPV), negative predictive value (NPV), accuracy, F1 score, and Matthews Correlation Coefficient (MCC). The F1 score is the harmonic mean between a tests PPV and sensitivity and ranges between 0 (worst performance) and 1 (perfect performance). To score highly a test must have both a high PPV and should miss very few cases (high sensitivity). The MCC is a correlation coefficient between predictions and outcomes and can be interpreted analogously to the Pearson's product moment correlation coefficient [23]. It incorporates all domains of the contingency table and therefore all classifier outcomes contribute equally to its calculation (i.e. misclassification of both positive and negative cases are important to its calculation). It ranges between -1 (test is always wrong) to +1 (test is always correct) with 0 indicating a performance no better than chance. Although our initial protocol dictated that we would attempt to generate summary receiver operator characteristic curves (SROC) their calculation demonstrated significant data heterogeneity that would render their findings invalid and, as such, were not included in this article.

# Results

## Study characteristics

Our initial search identified 534 records for screening with a further 24 identified through citation searching: These constituted 490 unique records. In total, 28 studies were included for qualitative review, from which classifier values could be extracted from 25. Our search was updated on two occasions. Initially, four new studies were included for qualitative analysis, including one paper where its inclusion proceeded after contacting the authors for unpublished data [24]. One of these studies was included in classifier analysis. A repeat search prior to publication identified four further studies from one of which measures of classifier performance could be calculated [25]. In total we therefore present results from 36 studies, with classifier values calculated from 27. Full inclusion and exclusion information is demonstrated in Fig 1. Studies covered multiple surgical specialities. Nine studies were performed in colorectal patients, six in vascular, six in upper gastrointestinal, six in hepatobiliary, six in mixed 'major surgical populations', two in urological, and one each in transplant and bariatric populations. Full details of all studies is shown in Table 1. The median number of study participants was 105 [Interquartile range 72-199]. 10 (28%) of studies were blinded meaning in the majority of studies (26/36–72%) information from CPET was either definitely or potentially used to guide the care of study participants (e.g. by stratifying admission to intensive care). In one study [26], despite not being blinded it was stated that CPET did not directly influence care. 23 (64%) of studies were performed prospectively. 28 studies looked at mortality as an outcome with 14 studies reporting on risk of combined cardiorespiratory morbidity. six studies reported purely cardiovascular complications, four respiratory complications, and six studies looked at rates of unplanned ICU admission. Risk of bias assessment for included studies, performed using the QUIPS checklist is shown in S1 Fig.

Selection of patients for CPET testing varied between studies. In two cases patients were selected for CPET on the basis of low levels of estimated functional capacity on a validated questionnaire. 12 studies used an age threshold for gating access to CPET, in seven of these a lower age threshold was used if the individual suffered from specific comorbidities (normally cardiovascular or diabetes). In five cases [27–31] we felt it was impossible to determine, due to phrasing of the papers or study design, exactly who was selected to undergo CPET and whether eligible participants might have been missed. In one study the inclusion criteria changed throughout the inclusion period from being selected on the basis of comorbidity to being consecutive. For details see Table 1.

## CPET variables and statistical significance

Studies looked at a variety of CPET derived variables. 33 studies (92%) looked at AT, 11 (31%) VE/VCO$_2$, 11 (31%) the Peak VO$_2$, and five (15.6%) VO$_2$ Max. Numerous other variables including cardiac ischaemia, oxygen pulse, work rates, chronotropic incompetence, and combined metrics were looked at by smaller numbers of studies. Details of all CPET variables studied in individual papers is shown in Table 1. No variable was found to be consistently statistically significant between studies. 17 of 33 studies (51.5%) found a significant association between AT and a pre-specified outcome of interest, 5/11 (45.4%) found associations between Peak VO$_2$, 4/11 (36.4%) VE/VCO$_2$, and 2/5 (40%) VO$_2$ Max. Of note, despite all studies utilising either a retrospective or prospective cohort design only one study utilised measures of relative risk [32]. None utilised Poisson or time-to-event statistical methods.

**Table 1. Summary of included studies in systematic review of the performance of cardiopulmonary exercise testing (CPET) for identification of post-operative outcomes of interest _n = 33_.**

| Study (Year) [Ref] | N | Specialty | Outcomes | Blinded | Prospective | Altered care | Selection for CPET | Variables | Statistics |
|---|---|---|---|---|---|---|---|---|---|
| Older (1993) [5] | 187 | Major Surgery | Mortality _(Total and CVS)_ | N | Y | Y | Consecutive age 60+ | AT* Ischaemia* | Chi-Sq |
| Nugent (1998) [44] | 36 | Vascular:AAA | Cardiorespiratory Mortality | Y | Y | N | Patients a/w repair _Excl. MSK or IC_ | AT Peak $VO_2$ $O_2$ Pulse Ischaemia | M-W, Chi-Sq |
| Older(1999) [6] | 548 | Major Abdominal | Mortality _(CVS and Other)_ | N | Y | Y | >60 <60 + previous Myocardial ischaemia. _Excl. Thoracic (incl. UGI)_ | AT | Descriptive |
| Nagamatsu (2001) [37] | 91 | Oesophagectomy | Cardiorespiratory | N | N | Unclear | Surgery for SCC _Excl. Neo-adjuvant chemotherapy, CCF, COPD_ | $VO_2$ Max* AT | T-T M-W |
| McCullough (2006) [45] | 109 | Bariatric | Mortality Cardiorespiratory _(MI, Angina, PE)_ | Y | Y | N | BMI ≥35 (+DM) or ≥40 for lap. roux-en-Y bypass. _Excl. Limiting angina, lung disease_ | Peak $VO_2$ | Descriptive (excluding analysis for composite outcome of death and complications) |
| Bowles(2008) [28] | 124 | Colorectal | Mortality | N | N | Y | Over 60 with 'recognised heart/ lung disease', post anaesthetic assessment | AT | Log. Regression |
| Forshaw(2008) [46] | 78 | Oesophagectomy | Mortality Unplanned ICU Cardiopulmonary | N | Y | Unclear | Consecutive | AT $VO_2$ peak* | T-T Fisher's |
| Struthers(2008) [47] | 50 | Major abdominal | Mortality(30D) | N | Y | Possibly | Over 65 having 'major intra-abdominal surgery' Under 65 with significant myocardial ischaemia, respiratory disease, or CCF. _Excl. unstable CVS disease_ | AT | Sens/Spec _(calculated)_ |
| Snowden (2010) [48] | 123 | Major abdominal | Mortality Cardiopulmonary | N | Y | Y | For 'major surgery' with METS score ≤ _Excl. Colorectal, Urological, and Orthopaedic surgery_ | AT* | Chi-sq |
| Wilson(2010) [32] | 847 | Colorectal Urology | Mortality | N | N | Y | >55 <55 + cardiorespiratory comorbidity or DM | AT* VE/VCO$_2$ * | RR |
| Thompson (2011) [49] | 66 | Vascular | Mortality(30D) Cardiac (+ Stroke), Respiratory | N | N | Y | All patients | AT* VE/VCO$_2$ VE/VO$_2$ * | OR (incl. non operative) _AT sig. mortality VE/VO$_2$ sig. inotropic requirement_ |
| Ausania (2012) [26] | 124 | Hepatobiliary | Mortality Cardiorespiratory | N | Y | N | Patients scoring <7 on a MET score | AT | Chi-sq |
| Hartley (2012) [29] | 415 | Vascular | Mortality(30D) | N | Y | Unclear | ?All undergoing open/EVAR | AT* VE/VCO2* ≥2 abnormal* Peak VO2* | Chi-sq |

_(Continued)_

**Table 1.** (Continued)

| Study (Year) [Ref] | N | Specialty | Outcomes | Blinded | Prospective | Altered care | Selection for CPET | Variables | Statistics |
|---|---|---|---|---|---|---|---|---|---|
| Junejo [33] (2012) | 94 | Hepatic | Mortality (30D+in-hospital) Cardiorespiratory | N | Y | Y | >65 <65 'with comorbidity', or 'complex resection' | AT* VE/VCO2* | OR *AT sig. in-hosp mortality, VE/VCO$_2$ sig. cardiorespiratory only* |
| Chandrabalan [50] (2013) | 100 | Pancreatic | Mortality *(incl. operative)*, Cardiac, Respiratory | N | N | Y | Patients for pancreatic surgery | AT | Chi-sq |
| Goodyear (2013) [30] | 85 | Vascular:AAA *Infrarenal* | Mortality(30D) | N | N | Y | Consecutive (not all) patients | AT* | Fisher's |
| Lai (2013) [51] | 269 | Colorectal | Mortality(30D) Unplanned ICU | N | Y | Y | All major colorectal | AT* "Unable to achieve AT" | Chi-sq |
| Moyes (2013) [52] | 180 | Upper GI | Mortality Cardiorespiratory Unplanned ICU | N | Y | Y | Consecutive after MDT discussion | AT* VO$_2$ peak | T-T *(AT sig. for Cardiorespiratory only)* |
| Prentis (2013) [53] | 69 | Cystectomy | Mortality | Y | Y | N | Consecutive | AT | Descriptive |
| Snowden[35] (2013) | 389 | Hepatobiliary | Mortality | Y | Y | N | All for open resection | AT* Peak VO2* Peak Work* VE/VCO2 VO2/HR | Chi-sq |
| Ting(2013) [34] | 70 | Renal Transplant | Unplanned ICU | Y | Y | N | All >18 *Excl.'condition precluding exercise'* | AT* PeakVO$_2$ * O$_2$ Pulse* Max Work* Endurance time VE/VCO$_2$ slope | Log. regression |
| Dunne (2014) [24] | 197 | Hepatectomy | Mortality *In hospital* Cardiorespiratory | N | N | Y | Initially >65 + 'significant comorbidity', or extended op. Changed to all patients | AT Peak VO$_2$ VE/VCO$_2$ VE/VO2 | Calculated% |
| James(2014) [39] | 83 | Major surgery | Mortality Cardiac | Outcome | Y | Possibly | Over 40s | AT* VO$_2$ Peak* | ?T-T |
| Junejo (2014) [38] | 64 | Pancreatic | Mortality Cardiorespiratory | N | Y | Unclear | >65 Younger with comorbidity | AT VE/VCO$_2$ * VO$_2$Max | Log. regression *Mortality only* |
| West(BJA 2014) [54] | 136 | Colorectal | Cardiorespiratory (D5) | Y | Y | N | All over 18 excl: neoadjuvant, IBD, inability to perform | AT*, | Chi-Sq Fisher's |
| West (EJSO 2014) [55] | 25 | Colorectal *(post NACRT)* | Cardiorespiratory | Y | Y | N | Surgery post NACRT —main aim to assess impact of NACRT on fitness | AT | Sens/Spec *(calculated)* |
| Barakat(2015) [27] | 130 | Vascular | Mortality Cardiac Respiratory | N | Y | Y | Most patients with AAA>55, able to use treadmill | AT* VO$_2$ Max VE/VCO$_2$ * | Log. regression *AT sig. cardiac only VE/ VCO$_2$ sig. resp. only* |
| Chan (2015) [56] | 94 | Colorectal | Mortality(30D), Unplanned ICU | N | N | Y | Subset of patients >80 at surgeons discretion | AT* VO$_2$ Max* | MW *(sig. ICU only)* |
| Nikolopoulous (2015) [57] | 89 | Colorectal | Cardirespiratory Mortality | N | N | Unclear | Consecutive (Open procedures) | AT* | T-T M-W *(Cardioresp. only)* |

*(Continued)*

**Table 1.** (Continued)

| Study (Year) [Ref] | N | Specialty | Outcomes | Blinded | Prospective | Altered care | Selection for CPET | Variables | Statistics |
|---|---|---|---|---|---|---|---|---|---|
| West [58] (2016) | 703 | Colorectal | Mortality *(30D+In hosp.)* Cardiorespiratory *(D5)* | N | Y | Y | Patients for major colorectal, excl: lower limb dysfunction, IBD, neoadjuvant treatment, metastatic | AT* | Fisher's |
| Kanakaraj (2017) [59] | 70 | Vascular *Peripheral* | Mortality(30D), Cardiac | Y | Y | N | Those for elective/ expedited infra-inguinal bypass surgery | AT Peak VO$_2$ VE/VCO$_2$ | T-T M-W |
| Whibley (2018) [31] | 73 | Upper GI | Respiratory | N | N | Unclear | Subset of those pre-assessed, part of an enhanced recovery protocol | AT VO$_2$ Max | Chi-sq |
| Abbott (2019) [41] | 1324 | Major non-cardiac | Myocardial injury D1-3 | Y | Y | Safety only | Over 70 OR Over 40 with higher risk surgery or comorbidity | Chronotropic Incompetence | Log regression |
| Drummond [25] (2019) | 42 | Oesophagectomy | Cardiorespiratory, 30D Mortality, Unplanned ICU | N | Y | Unclear | Selected patients with pre/post chemotherapy prior to oesophagectomy for adenocarcinoma | AT (Pre/Post Chemotherapy) | Chi-sq |
| Lam (2019) [36] | 206 | Oesophagectomy | Cardiorespiratory | Y | N | Possibly | Consecutive undergoing oesophagectomy for cancer. Excluded 40 who didn't undergo CPET for unclear reasons | AT Peak VO$_2$ | T-T |
| Wilson (2019) [60] | 1375 | Colorectal | Unplanned ICU | N | N | Y | Over 55 or younger with cardiorespiratory risk factors | VE/VCO$_2$ | Log regression |

30D = 30 Day, AAA = Abdominal aortic aneurysm, AT = Anaerobic Threshold, BMI = Body Mass Index, CCF = Congestive Cardiac Failure, Chi-Sq = Chi-Square test, COPD = Chronic Obstructive Pulmonary Disease. CPET = Cardiopuulmonary Exercise Testing, CVS = Cardiovascular system, DM = Diabetes Mellitus, EVAR = Endovascular Aneurysm Repair, IC = Intermittent claudication, ICU = Intensive care unit, lap. = laparoscopic, METS = Metabolic equivalents, MDT = Multi-disciplinary Team, MI = Myocardial Infarction, N = number of patients who underwent and/or were analysed as having had CPET and may differ from the total number of patients included in some studies, NACRT = Neoadjuvant chemo-radiotherapy, Chi-Sq = Chi- squared test, M-W = Mann-Whitney test, MSK = musculoskeletal pathology, OR = Odd's Ratio, PE = Pulmonary embolism, RR = Relative Risk, SCC = Squamous cell carcinoma, Sens/Spec = Sensitivity/Specificity, T-T = Student's T-Test, VE/VCO$_2$, VE/VO$_2$ = Ventilatory equivalents of carbon dioxide/oxygen, VO$_2$ Max = Maximal oxygen uptake. Where stated D refers to postoperative day. *Italics indicate supporting information or exclusion criteria*

* next to a CPET variable indicates it was found to be significant (using methods in statistical methods column.

'Unclear' was used to indicate if it was not possible to determine whether CPET values could have impacted on any aspect of patient care and thus introduced confounding. Statistical methods/CPET variables shown are those pertaining to our specified outcomes of interest only. 'Sens/Spec (calculated)' demonstrates that we extracted data pertaining to these CPET variables and outcomes and no formal testing was presented within the paper. % Indicates that analysis was performed on data requested from the authors after not being presented in the primary paper.

## Prediction performance metrics

27 studies either presented sensitivity and specificity or presented data in a format that allowed us to calculate these and other performance measures by generating confusion matrices. These studies, values, outcomes, and CPET cut-offs are presented in Table 2. Due to the multitude of surgical specialities, outcome definitions, and differences in defined populations who underwent CPET we did not generate 'average' values of each metric for each outcome. Instead this

**Table 2. Confusion matrix metrics for CPET derived cardiorespiratory variables.**

| Study | Outcome | CPET Variable/Cutoff | Sensitivity | Specificity | PPV | NPV | Accuracy | F1 | MCC |
|---|---|---|---|---|---|---|---|---|---|
| Chan [56] | Mortality | CPET Group | 20.0 | 47.2 | 2.1 | 91.3 | 45.7 | 0.04 | -0.15 |
| Hartley [29] | 30D Mortality | VE/VCO$_2$ 42 | 42.9 | 80.5 | 7.1 | 97.6 | 79.2 | 0.12 | 0.11 |
| Hartley [29] | 30D Mortality | Peak VO$_2$ 15 | 85.7 | 47.9 | 5.4 | 98.9 | 49.2 | 0.10 | 0.12 |
| Hartley [29] | 30D Mortality | 2 or more abnormal | 85.7 | 59.1 | 6.8 | 99.2 | 60.0 | 0.13 | 0.16 |
| Hartley [29] | 30D Mortality | AT 10.2 | 78.6 | 55.1 | 5.8 | 98.7 | 55.9 | 0.11 | |
| Goodyear [30] | 30D Mortality | AT<11 (or unable to achieve) | 50.0 | 88.9 | 18.2 | 97.3 | 87.1 | 0.27 | 0.25 |
| Older [5] 1993 | IP CVS Mortality | AT 11 | 90.1 | 74.4 | 18.2 | 99.2 | 75.4 | 0.30 | 0.34 |
| Older [5] 1993 | IP CVS Mortality | AT 11 + Ischaemia | 88.9 | 68.6 | 42.1 | 96.0 | 72.7 | 0.57 | 0.47 |
| Older [5] 1993 | IP CVS Mortality | Ischaemia | 81.8 | 80.1 | 20.5 | 98.6 | 80.2 | 0.33 | 0.34 |
| Wilson [32] | In-hospital mortality | AT 10.9 | 88.9 | 46.8 | 3.5 | 99.5 | 47.7 | 0.07 | 0.10 |
| Wilson [32] | In-hospital mortality | VE/VCO$_2$ 34 | 83.3 | 48.5 | 3.4 | 99.3 | 49.2 | 0.07 | 0.09 |
| Wilson [32] | In-hospital mortality | VE/VCO$_2$ 42 | 27.8 | 88.5 | 5.0 | 98.3 | 87.2 | 0.08 | 0.07 |
| Ausania [26] | In-hospital mortality | AT 10.1 | 20.0 | 84.0 | 5.0 | 96.2 | 81.4 | 0.08 | 0.02 |
| Junejo [33] | 30D mortality | AT 9.35 | 67.0 | 83.0 | - | - | - | - | - |
| Junejo [33] | 30D mortality | AT 9.9 | 100 | 74.0 | - | - | - | - | - |
| Junejo [33] | In Hosp mortality | AT 9.35.0 | 40.0 | 83.0 | - | - | - | - | - |
| Junejo [33] | In Hosp mortality | AT 9.9 | 100 | 76.0 | - | - | - | - | - |
| Junejo[38] | 30D mortality | VE/VCO$_2$ 41 | 100 | 92.0 | - | - | - | - | - |
| Junejo [38] | In Hosp mortality | VE/VCO$_2$ 41 | 75.0 | 93.0 | - | - | - | - | - |
| Lai [51] | 30D Mortality | AT 11 | 40.0 | 71.8 | 2.9 | 98.3 | 71.2 | 0.05 | 0.04 |
| Lai [51] | 30D Mortality | Unable to get AT | 44.4 | 91.5 | 15.4 | 97.9 | 90.0 | 0.23 | 0.22 |
| McCullough [45] | Mortality | Peak VO$_2$ 15.8 | 100 | 66.7 | 2.7 | 100 | 67.0 | 0.05 | 0.13 |
| Struthers [47] | Mortality(30D) | AT 11 | 100 | 68.4 | 7.7 | 100 | 69.2 | 0.14 | 0.22 |
| Prentis [53] | Mortality | AT 12 | 100 | 58.2 | 6.7 | 100 | 59.4 | 0.13 | 20 |
| Chandrabalan [50] | Mortality | AT 10.0 | 42.9 | 50.5 | 6.1 | 92.2 | 50.0 | 0.11 | -0.03 |
| Nugent [44] | Mortality | Peak VO$_2$ 20 | 100 | 44.8 | 5.9 | 100 | 46.7 | 0.11 | 0.16 |
| Drummond [25] | Operative mortality | AT 9 | 28.6 | 82.1 | 44.4 | 69.7 | 64.3 | 0.35 | 0.12 |
| Drummond [25] | Operative mortality | AT 11 | 0 | 41.5 | 0 | 94.4 | 40.5 | 0 | -0.18 |
| James [39] | MACE | AT 10.6 | 75.0 | 85.0 | - | - | - | - | - |
| James [39] | MACE | Peak VO$_2$ 14 | 88.0 | 69.0 | - | - | - | - | - |
| Snowden [48] 2010 | Cardiac | AT 10.1 | 86.7 | 62.4 | 25.5 | 96.9 | 65.5 | 0.39 | 0.33 |
| West [54] BJA | MI/Arrhythmia | AT 10.1 | 64.7 | 55.4 | 17.2 | 91.7 | 56.6 | 0.27 | 0.13 |
| West [55] EJSO | Arrhythmia | AT 10.7 | 100 | 50.0 | 7.7 | 100 | 52.0 | 0.14 | 0.20 |
| Chandrabalan [50] | Cardiac (CD III-IV) | AT 10 | 100 | 51.5 | 2.0 | 100 | 52 | 0.04 | 0.10 |
| West [58] | Cardiovascular | AT 11.1 | 72.2 | 60.4 | 4.6 | 98.8 | 60.7 | 0.09 | 0.11 |
| Ausania [26] | Cardiorespiratory | AT 10.1 | 33.3 | 85.2 | 15.0 | 94.2 | 81.4 | 0.21 | 0.13 |
| Forshaw [46] | Cardiorespiratory | AT 11 | 23.3 | 88.9 | 58.3 | 63.5 | 62.7 | 0.33 | 0.16 |
| Moyes [52] | Cardiorespiratory | AT 9 | 37.9(45) | 79.7(30) | 42.3 | 76.6 | 68.0 | 0.4 | 0.18 |
| Moyes [52] | Cardiorespiratory | AT 11 | 69.0(74) | 50.0(57) | 35.1 | 80.4 | 55.3 | 0.47 | 0.17 |
| Nikolopolous [57] | Cardiorespiratory | AT 11 | 76.0 | 59.0 | - | - | - | - | - |
| Nagamatsu [37] | Cardiorespiratory | VO$_2$ Max 800 | 58.8 | 91.9 | 62.5 | 90.7 | 85.7 | 0.61 | 0.92 |
| Junejo [38] | Cardiorespiratory | VE/VCO$_2$ 34.5 | 50.0 | 81.5 | - | - | - | - | - |
| Junejo [33] | Cardiorespiratory | VE/VCO$_2$ 36.5 | 39.5 | 90.7 | - | - | - | - | - |
| McCullough [45] | Cardiorespiratory | Peak VO$_2$ 15.8 | 50.0 | 66.7 | 5.4 | 97.2 | 66.2 | 0.10 | 0.07 |
| Nugent [44] | Cardiorespiratory | PeakVO$_2$20 | 57.1 | 43.4 | 23.5 | 76.9 | 46.7 | 0.33 | 0.01 |
| Drummond [25] | Cardiorespiratory | AT 9 | 38.9 | 83.3 | 63.6 | 64.5 | 64.3 | 0.48 | 0.25 |
| Drummond [25] | Cardiorespiratory | AT 11 | 61.1 | 45.8 | 45.8 | 61.1 | 52.4 | 0.52 | 0.07 |
| West [54] BJA | Pneumonia | AT 10.1 | 83.3 | 56.5 | 15.6 | 97.2 | 58.9 | 0.26 | 0.23 |

(*Continued*)

**Table 2.** (Continued)

| Study | Outcome | CPET Variable/Cutoff | Sensitivity | Specificity | PPV | NPV | Accuracy | F1 | MCC |
|---|---|---|---|---|---|---|---|---|---|
| West [55] EJSO | Pneumonia | AT 10.7 | 100 | 52.2 | 15.4 | 100 | 56.0 | 0.27 | 0.28 |
| Snowden [48] 2010 | Pulmonary | AT 10.1 | 74.4 | 71.4 | 56.9 | 84.6 | 72.4 | 0.64 | 0.44 |
| Chandrabalan [50] | Respiratory (CD III-IV) | AT 10 | 57.1 | 51.6 | 8.2 | 94.1 | 52.0 | 0.14 | 0.04 |
| West [58] | Respiratory | AT 11.1 | 76.3 | 64.2 | 21.4 | 95.5 | 65.6 | 0.34 | 0.26 |
| Ting [34] | Unplanned ICU | AT 11.3 | 93.0 | 75.0 | - | - | - | - | - |
| Chan [56] | Unplanned ICU | CPET group | 33.3 | 64.3 | 16.7 | 81.8 | 58.8 | 0.22 | -0.02 |
| Forshaw [46] | Unplanned ICU | AT 11 | 23.1 | 84.1 | 25.0 | 85.4 | 74.7 | 0.24 | 0.09 |
| Lai [51] | Unplanned ICU | AT 11 | 39.1 | 72.3 | 13.0 | 92.0 | 69.5 | 0.20 | 0.08 |
| Prentis [53] | Unplanned ICU | AT 12 | 60.0 | 70.6 | 16.7 | 94.7 | 69.6 | 0.26 | 0.19 |
| Drummond [25] | Unplanned ICU | AT 9 | 35.7 | 78.6 | 45.4 | 71.0 | 64.3 | 0.40 | 0.15 |
| Drummond [25] | Unplanned ICU | AT 11 | 71.4 | 50.0 | 41.7 | 77.8 | 57.1 | 0.53 | 0.20 |

Outcome refers to study specified outcome of interest, CPET variable and cutpoint for values presented. Units for cutoffs: AT = ml/kg/min, Peak $VO_2$ = ml/kg/min, $VO_2$ Max = ml/min/m$^2$, VE/$VCO_2$ = Slope of curve between minute ventilation and $CO_2$ production. PPV = Positive predictive value, NPV = Negative predictive value, F1 = F1 Score, MCC = Matthew's Correlation Coefficient, CD = Clavien Dindo classification of complications. AT = Anaerobic Threshold, VE/$VCO_2$ = Ventilatory equivalents of carbon dioxide, Peak $VO_2$ = highest recorded oxygen uptake, $VO_2$ Max = Maximal $VO_2$ uptake, 30D Mortality = 30 Day mortality, MACE = Major adverse cardiovascular events, IP CVS Mortality = Inpatient deaths from cardiac causes, ICU = Intensive Care Unit. If only sensitivity/specificity shown then these are published figures extracted from the paper. Figures in parentheses show published values for sensitivity and specificity, mismatch to calculated—potentially due to double counting of complications due to cutpoints used in published tables. Figures for unplanned ICU admission for Prentis excludes patients electively admitted to ICU.

table summarises ranges based on the outcomes of: mortality (all timescales up-to 30 days), cardiac complications alone, respiratory complications alone, combined morbidity, and unplanned ICU admission.

The most complete information was available for AT. For mortality, sensitivity values for AT ranged from 0-100%, specificity ranged from 41.5% to 92%. NPV of AT for mortality had a high estimated range (94.4-100%) which was higher than that for PPV (0–44.4%). These values are presented graphically in the boxplot in Fig 2 along with corresponding ranges for cardiorespiratory complications, and unplanned ICU admission. The range for MCC values was -0.18-0.47. F1 values ranged between 0 and 0.57.

For Peak $VO_2$ as a predictor of mortality the sensitivity was consistently high ((85.7% to 100% from three studies) with correspondingly high NPV figures (98.9%–100%). Specificity ranged from 44.8% to 66.7%. F1 scores ranged between 0.05 and 0.11 (reflecting PPV values of 2.7% to 5.9%), MCC was between 0.12 and 0.16.

VE/$VCO_2$ demonstrated a wide range of sensitivities for predicting mortality ((27.8% to 100%)) with specificity of 48.5% to 93.0%. NPV was consistently above 97% and PPV between 3.4% and 7.1%. All MCC, and F1 values were 0.12 or lower. Data from more than one study was available for Peak $VO_2$, and VE/$VCO_2$ as predictors of mortality, these are graphically demonstrated in S2 Fig. Values for other CPET variables are shown in Table 2.

## Logistic regression models and CHARMS checklist

Seven studies generated a multivariable model for one of our pre-specified outcomes using, or including, CPET derived variables (Table 3). We critiqued these studies using the CHARMS checklist. All were derived from Cohort studies, four of which were prospective [27, 33–35], two were retrospective [36, 37], and in one case it was unclear [29]. Two studies generated a model to predict mortality, four cardiorespiratory complications, and one unplanned ICU admission. The populations from which the models were generated varied by surgical

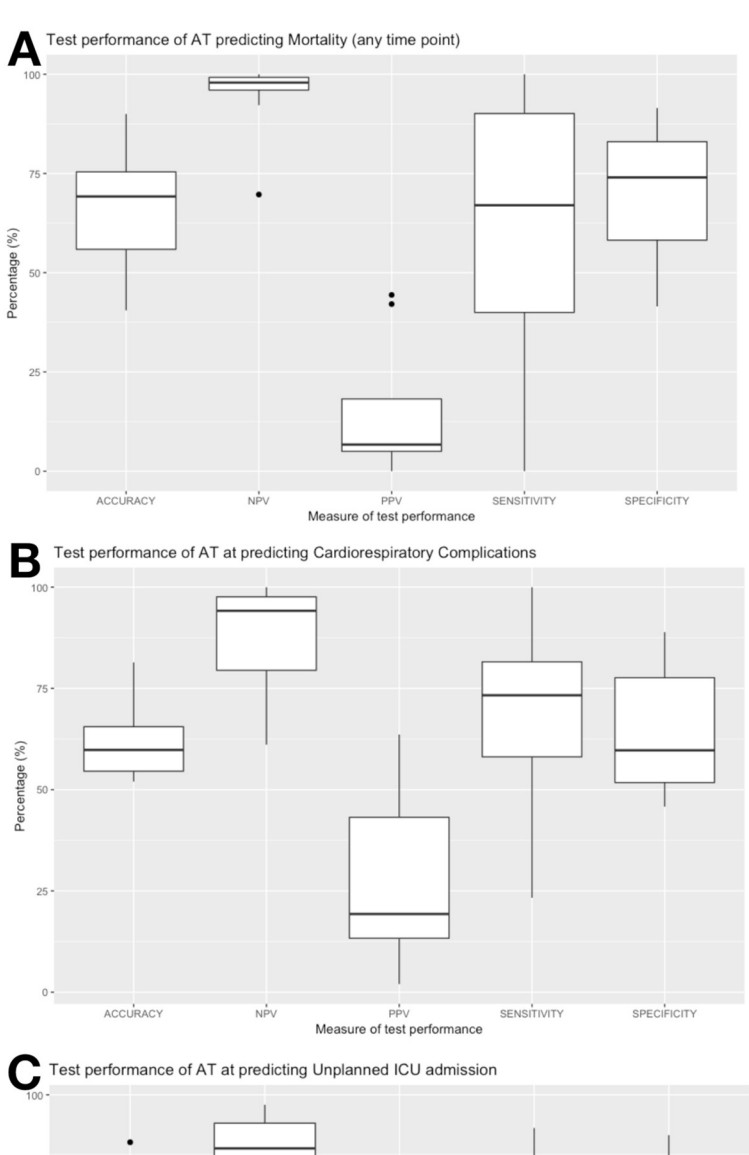

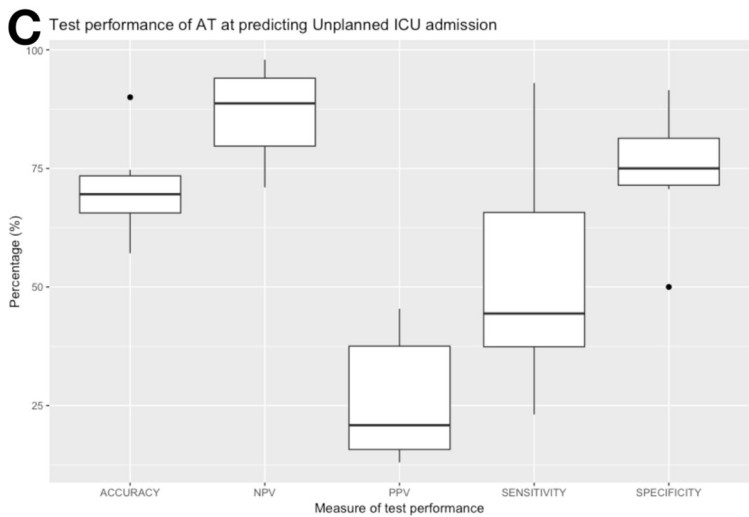

**Fig 2. Boxplots of calculated and published measures of test performance of anaerobic threshold in identification of individuals who suffer specific post-operative complications (n = 17 Studies).** Results are across heterogeneous surgical populations and use a variety of cut-offs and outcome definitions. A: Performance for identification of Mortality (up-to 30 day postoperatively), B: Cardiorespiratory complications (alone and/or in combination), C: Unplanned ICU admission. For source data see Table 2.

**Table 3. Assessment of prediction models in identified studies using the CHARMS checklist (CHecklist for critical appraisal and data extraction for systematic reviews of prediction modelling studies.**

| Study | Source | Participants | Outcomes | Predictors | Development | Sample Size (EVR) | Missing Data | Performance | Evaluation |
|-------|--------|-------------|----------|-----------|-------------|-------------------|--------------|-------------|-----------|
| Nagamatsu [37] | Cohort (R) | Individuals undergoing oesophagectomy 97% M | Cardiorespiratory | $VO_2$ max AT VC $FEV_1$, V50/V25 ratio, DLCO, $PaO_2$ | Multiple regression, unclear model building strategy | 91 (2.4) | N/A | ROC *No AUC* | Not done |
| Hartley [29] | Cohort (R/P) | 84%M, 23% >80 undergoing open and EVAR AAA repair | Mortality(30D) | Repair type Sex, Age, DM, IHD, HTN, Antiplatelets, Statins, Anaemia, Urea, Creatinine AAA size & location, AT, Peak $VO_2$, VE/VCO$_2$ | Univariable prior to Multi, backwards using AIC | 451 (3.5) | Excl. if >15% missing Median imputation for continuous classed as absent if unclear for categorical | Not done | Not done |
| Junejo [33] | Cohort (P) | Undergoing hepatic resection >65 or with complex resection, average age 71, 2:1M:F | Cardiorespiratory | Age ASA BMI RCRI AT VE/CO$_2$, | Unclear. *P* threshold for progression to Multiple Regression analysis of <0.1 mentioned | 92 (19.5) | N/A | Not done | Not done |
| Ting [34] | Cohort (P) | Kidney transplant, average age 42, 91% living kidney recipients, 60% Male | Unplanned ICU | BMI Desensitisation Age, Sex Months of Dialysis, HTN, Dyslipidemia, DM Carotid-Femoral PWV (m/s) Augmentation index AT | Multivariable after univariable *P* of <0.10, FW and BW step. BMI and dialysis time incl. as known confounders of postop cardiovascular state. AT used as binary value (11.3) | 70 (4.7) | N/A | AUC HL test | Cross-validation *(no details)* |
| Snowden [35] | Cohort (P) | Consecutive for open hepatic resection, excluded those unable to obtain an AT | Mortality | AT Peak $VO_2$ Peak Work Peak HR, Age, BMI, Sex IHD, CVA, DM, CCF, HTN, Asthma, COPD RCRI | Univariable then included those with *P*<0.1, forward step regression with assessment of co-linearity. Only present final model containing AT alone. | 389 (18) | N/A | AUC | Not done |
| Barakat [27] | Cohort (P) | Single centre study, Male preponderance (89%), Unclear percentage of eligible population sent for CPET | Cardiac Respiratory | Age, Sex Method of repair AT $VO_2$ peak VE/VCO$_2$ | Included set variables, no evidence of progression from univariable to multivariable. No process of model simplification given. | 130 (R = 4) (C = 3) | Not stated | Not done | Not done |
| Lam [36] | Cohort (R) | Single centre, aimed for all consecutive patients undergoing oesophagectomy for cancer | Cardiorespiratory | AT Peak $VO_2$ Type of Surgery Gender Comorbiditty Smoking Stage and histology Adjuvant therapy | Included CPET variables and all with univariable *p* of <0.10 | 80 (20) | Not stated | Not done | Not done |

speciality (Upper GI, vascular, hepatobiliary, renal transplant) and, in one paper [27], an unclear percentage of the eligible population were sent for CPET. Included covariates included CPET variables (AT, VE/VCO$_2$, VO$_2$ peak, work parameters) as well as scoring systems (ASA score, revised cardiac risk index), comorbidites, and specific medications. In four cases a clear process of model development was followed with the inclusion of variables significant on univariable analysis (normally $p<0.1$) or for pre-specified biological reasons, being included in initial multivariable models [29, 34–36]. At the end of model generation one study [35] simplified to using AT alone as a predictor variable, and in one case [27] the process of model generation or simplification was unclear. Median sample size was 123 [IQR: 93-168]. Three of the studies had event to variable ratios (EVR) of above 10. An EVR of less than 10 is associated with a risk of 'overfitting' and generating a model with poor external validity [19]. Only two of the papers reported an assessment of model performance. One reported tests of discrimination (AUC) and calibration (Hosmer-Lemeshow) [34] with one reporting only a measure of discrimination (AUC) [35]. Only one study reported to have undertaken a process of internal evaluation of model performance (unspecified cross-validation approach) [34], no model was verified in an external dataset.

### AUC and other predictors

Nine studies reported AUC values for CPET variables. For mortality these ranged from 0.75 (AT) to 0.95 (VE/VCO$_2$). Presented values for cardiorespiratory complications appeared lower and ranged from 0.54 (Peak VO$_2$) to 0.83 (AT). Full values are available in S1 Table. For seven studies classifier measures for other predictors (e.g. ASA score) were presented, or calculated (S2 Table). Potential predictors included blood tests, other scoring systems, or clinician determined high and low risk groups. For those studies looking at mortality [26, 28, 32, 33, 38], sensitivity ranged from 44.4% (presence of a single cardiac risk factor) to 100% (study delineated 'high risk group' based on CPET referral criteria). In the paper by James *et al* [39]. AUCs for a variety of other predictors were generated but no tests of significance between these and the AUCs calculated for CPET variables was performed.

## Discussion

To our knowledge, this is the first systematic review to extract and calculate comparable measures of CPET's performance as a prognostic test. We feel this framing of CPETs performance in familiar numerical terms is a key strength of our study as it provides a clear structure for considering the implications of CPET's performance in routine practice.

A key paper in the field was discovered during our repeat search [40]. This was a prospective trial of over 1300 patients in which CPET was compared to both subjective clinician assessment and other objective methods of physical performance. Due to its importance we reviewed it for inclusion in our systematic review but it failed to meet our inclusion criteria due to its use of a pooled outcome of 30 day death and myocardial infarction. We will however highlight key messages from this paper in the context of our other findings. We did include a secondary analysis of this cohort where chronotropic incompetence during CPET was assessed as a predictor of postoperative myocardial injury assessed using rises in serum troponin [41].

### Classifier performance of CPET

The differences in calculated NPV and PPV across studies examining AT is particularly striking (Fig 2). Both NPV and PPV depend on the frequency of the complication within the population, [42] being at their most extreme with a rare outcome (such as mortality). A gradual reduction in NPV, and increase in PPV as complications become more frequent (e.g. cardiorespiratory complications) can be seen. We also generated two lesser known measure of classifier

performance, the MCC and the F1 score [15]. Its interpretation is relatively intuitive. It is a correlation coefficient, ranging from -1 to 1 with the two bounds representing a test that is always incorrect, and one that is always correct. Importantly, given the paucity of events in some studies, the measure is relatively resistant to class imbalance (i.e. low frequency of specific outcomes). We report a broad range (from -0.18 to 0.92) but with median values for our differing complications all being less than 0.17 indicating, across all domains of patient classification that CPET's performance is relatively poor. Although no single measure of a test's performance can fully encompass its clinical utility, in the context of CPET the F1 score arguably comes close. This is a score that equally weights the PPV and the sensitivity both of which, ideally, should be high (i.e. a positive CPET test indicates a high likelihood of developing a complication and that there are few false negatives). Across all outcomes and studies our estimates ranged from 0.04 to 0.64 with 22/48 values being 0.20 or lower. As shown in Table 2 studies with higher F1 scores demonstrated higher PPV. A deeper understanding of why these studies exhibited better PPV could be vital in determining optimal use of CPET (e.g. was it due to specific procedures, recruitment measures, or derived indices). The F1 score weights both false positives and false negatives equally but, depending on context this weighting can be adjusted (the F$\beta$) score) [15]. We utilised equal weighting in the CPET context as although we would like a sensitive measure, potential interventions (such as critical care beds), have a finite capacity and thus could not accommodate unlimited numbers of people who will actually not suffer a complication.

Our results suggest that most individuals with a 'high-risk' CPET result actually have a greater chance of not suffering a complication than they are to suffer one. This poor rule-in performance is important if decisions to not proceed to surgery are taken on the basis of such a result. It is also highly relevant if a scarce resource (such as a critical care bed) is being allocated preemptively based on a low likelihood of a complication occurring. Worse still, we have no systematic information to suggest whether the allocation of a critical care resource in the immediate post-operative period would be effective in preventing adverse events. Not only is this impossible from such observational work, but this may also be determined by timings of adverse events which may occur after a preemptive critical care admission is over.

Conversely, rule out performance is better, with a 'low-risk' CPET patient being unlikely to suffer complications meaning, in the correct context, this could reassure clinicians about proceeding with surgery. Acceptable thresholds for NPV, and PPV in terms of gating further care are open for discussion but should be considered in light of their potential benefits on the shared decision-making process in the run up-to surgery. Overall our results would suggest that CPET is better at ruling out complications (as indicated by its relatively high NPV) but this is also likely influenced by the frequency of complications and the sample size within the study in question. A nationwide survey in 2011 suggested that 33% of CPET centres may recommend cancellation based on CPET results, although in the majority of cases the results form part of the overall shared decision making process [9]. From our results we cannot comment specifically on CPET's performance in this context although each result that we have calculated will be dependent on the preceding stratification process which was highly variable between studies (Table 1). Given its complexity in terms of organisation, staffing, and patient attendance the justification of CPET over other stratification tools relies on a demonstration that it offers additional benefit to patients and clinicians. An argument could be made, that in a sufficiently 'high-risk' population the frequency of complications would increase to a point that the NPV, and PPV of CPET would become more favourable. The correct method of doing this is not apparent from our included studies. A cautionary note for such a strategy is the fact that, in a population selected on the basis of cardiac risk factors, CPET offered no additional discriminating capabilities on risk of cardiac complications or death [40].

These, and other gaps in the literature have only recently been filled by the prospective, multi-centre 'METS' (Measurement of Exercise Tolerance before Surgery) trial [40]. This followed nearly 1,400 patients for one year after major, non-cardiac surgery, with daily examination for cardiac complications whilst an inpatient. They utilised the 'net reclassification index' to identify whether CPET, when added to baseline models of age, sex, cardiac risk factors, and surgery type was able to correctly reclassify individuals at risk of complications. For their primary outcome of 30-day mortality or myocardial infarction they showed no additional prognostic benefit of CPET derived variables. The only significant CPET variable was the addition of peak oxygen consumption for the identification of those at risk of a composite of all moderate to severe in-hospital complications. The study also assessed both the Duke Activity Score Index (DASI), and NT-pro-BNP for improved identification of those at risk of their primary outcome. Neither were significant for this outcome but, both demonstrated benefit at identifying sub-clinical myocardial injury (as assessed by troponin assay). The METS study suggests that CPET adds little to easily assessed baseline factors for predicting risk. An important caveat might be the frequency of outcomes, with the study only having 28 individuals suffering their composite primary outcome [40].

In addition, our study is the first to apply the CHARMS [20] criteria to regression models generated using CPET variables. We critiqued models generated against any one of our pre-specified outcomes. In general, generation, and assessment of regression models was incomplete. Only one study assessed internal validity (using cross-validation) and three of the models were at risk of overfitting due to low event to variable (EVR) ratios. No model has (from our results) ever been validated in a different patient cohort or used as a starting point for refinement by subsequent authors. As such, generalising the findings of these analyses is unfounded. The lack of clear outputs of model performance (such as discrimination, and calibration) means that even internal validity may be difficult to determine in certain cases.

The results of our systematic review indicate a broad range of estimates for CPET performance as a prognostic test in non-cardiopulmonary surgery. The single centre, retrospective nature of many of the studies limits the ability to generalize findings to everyday practice and it is telling that the original paper identifying an AT of 11ml/kg/$O_2$ as an important threshold is now 25 years old and, arguably, may not be reflective of today's surgical practice. It is telling that, as highlighted in Table 1, many studies did not blind clinicians to the results of CPET, introducing the potential for bias and confounding into their (and, subsequently, our) findings. Nationally, CPET has been used to determine various components of perioperative care including ICU admission, monitoring, or type of surgical procedure [9]. Some of these interventions (e.g. routine critical care for high risk individuals) could, arguably, bias findings towards the null by having averted potential complications in the high-risk cohort. The fact that 7 of 11 studies where CPET was explicitly or potentially stated to have not altered care, failed to demonstrate significant associations with complications suggests this is unlikely to be the whole truth.

Whilst our work illustrates that CPET alone is not a good predictor of adverse events our results may not fully encompass the subtleties of its use within clinical practice. Certainly it is possible that there exists a spectrum of use and it may be that CPET is of predictive value in a subset of very high risk patients. Unfortunately, a lack of methodological standardization makes this impossible to assess. Indeed, in our included studies there has been little or no systematic attempt to assess the incremental discrimination of CPET over and above other, perhaps simpler assessments.

## Strengths and weaknesses of our approach

We acknowledge that due to the potential for bias and confounding within the primary studies that our extracted and calculated estimates are, themselves, likely to be biased. We defend this by arguing that the widespread adoption of CPET is arguably grounded in the interpretation of these studies and that we are merely quantifying metrics which should be implicit in the extrapolation of these studies to clinical practice. We judged that calculating classifier metrics based on these studies was a valid approach as we focused on a single time period (the in-hospital period) and that the majority of studies did not delineate the exact time a complication was screened for or occurred. If raw data was available then use of Poisson or time to event (e.g. Cox) regression techniques would be an alternative to demonstrate an association between CPET variables and outcomes of interest. We feel that our use of classifier measures in this prognostic context (which is supported in the literature [14]) offers a framework for interpretation that can be directly related to clinical practice.

Our results are heterogeneous due to the plethora of settings, surgical sub-specialities, selection criteria, outcomes, and outcome definitions used. It is for this reason that we have taken no steps to present a pooled or aggregate measure of CPET's performance. This is an important finding in itself and demonstrates a troubling lack of methodological standardisation in the literature and presumably therefore perhaps also in how CPET is used clinically. Our initial aim had been to only include studies utilising rigorously defined definitions of specific postoperative complications [21] but this had to be broadened to any named complication within this document to allow for a suitable number of studies to be included. We justify this by saying that any concerns we had with the chosen method of outcome definition is reflected in the QUIPS score given to the study for that domain (S1 Fig). Our selection of outcomes for inclusion was underpinned by physiological rationale and the utilisation of CPET in routine practice. We tried to minimise the use of composite outcomes to reduce heterogeneity in our generated classifier metrics. We did extract pooled cardiorespiratory outcomes if so presented due to the potential for crossover between these measures depending on the scoring system used. For instance the postoperative morbidity score requires an individual to be on oxygen to class as having a respiratory complication, which is highly likely to have occurred in the context of an individual with heart failure (who would be classed as having a cardiovascular complication) [21]. As a caveat to this, if multi domain scores like this were presented we calculated the occurrence of cardiac and respiratory complications separately to minimise the risk of double counting individuals. The use of composite outcomes is widespread especially as they can lead to an increase in statistical power [43]. Issues with this approach are that any effect may be interpreted by the reader to apply equally across all components of the composite, and that there should be an equivalent effect on each component. Furthermore, an interesting area of further work would be to explore how valid the use of 'all complications' is from a patient perspective, the presentation of information pertaining to such a composite is arguably only at its most valid if the patient weighs the impact of each component equally.

We accept that in many of the included studies our outcome was not the author's primary outcome and this may introduce bias into our results by skewing indices such as sensitivity and specificity (See calculations in S1 File). We hope that by providing full access to our calculations, including the location of generated numbers within the corresponding paper, that we have provided sufficient information for clinicians to interpret our findings in context.

## Conclusion

Our systematic review highlights potential limitations of CPET by generating common and novel measures of test performance. Specifically, given the frequency of complications in

studies we highlight the low values of PPV, although far from perfect for gating access to surgery or critical care they may help inform a process of shared decision making prior to surgery. The performance of CPET in day to day practice may be improved if a sufficiently high risk population could be identified for it to be employed in. The high NPV offered by CPET may be testament to accuracy of its underlying physiological rationale. However, based on our results, and those from a recent high-quality prospective study [40] the precision of CPET for reliably identifying individuals who will suffer postoperative cardiorespiratory complications appears poor. Certainly, given the values of PPV we report it would be difficult to justify not proceeding to surgery on the basis of a high-risk CPET test alone. Due to its role within the wider system of perioperative assessment, optimisation, and postoperative care a nuanced interpretation of its ability to rule-in and rule-out potential complications is justified.

## Supporting information

**S1 Appendix. Literature search strategy.** Example given for repeat search—same terms used for initial search, see Methods.
(PDF)

**S2 Appendix. PRISMA checklist.**
(DOC)

**S3 Appendix. PROSPERO registration form.**
(PDF)

**S1 Fig. Risk of bias assessment for included studies.** Scoring performed using the QUIPS (Quality in Prognosis Studies) tool. Reported values are those reached after individual assessment by two authors and arbitration amongst all three. Red = High risk of bias, Orange = Moderate risk of bias, Green = Low risk of bias (for associated domain).
(TIF)

**S2 Fig. Boxplots of calculated and published measures of test performance, and positive and negative predictive values for peak VO2(A), and VE/VCO (B) as predictors of mortality.** Source data can be seen in table of calculated confusion matrix metric (S1 File).
(TIF)

**S1 File. Confusion matrices.**
(XLSX)

**S1 Table. Published area under the receiver operator characteristic (ROC) curve values from identified studies.** Brackets indicate published 95% confidence interval. Asterisk indicates that published confidence interval crosses 1 (likely calculated as a normal appproximation).
(PDF)

**S2 Table. Performance of non cardiopulmonary exercise testing (CPET) variables for prediction of specific outcomes.** MET = Metabolic equivalents, BNP = B type natriuretic peptide, NT-proBNP = N-terminal pro b-type natriuretic peptide, eGFR = estimated glomerular filtration rate, CRP = C Reactive Protein, ASA = American Society of Anesthesiologists score, RF = Risk Factors, IHD = Ischaemic Heart Disease, AT = Anaerobic Threshold, RCRI = Revised Cardiac Risk Index. POSSUM = Physiological and Operative Severity Score for the Enumeration of Morbidity and Mortality. AUC = Area under the receiver operator curve, OR = Odds ratio, * calculations exclude 14 high risk patients who didn't proceed to surgery

based 'partly' on CPET findings.
(PDF)

## Acknowledgments

DJS is supported by a Wellcome Trust Clinician PhD Fellowship [Overarching grant: 204017/Z/16/Z].

The authors would like to thank Prof Simon Griffin and Dr Juliet Usher-Smith for numerous helpful discussions as well as Ms Isla Kuhn for assistance with the design of search strategy.

## Author Contributions

**Conceptualization:** Daniel J. Stubbs, Ari Ercole.

**Data curation:** Daniel J. Stubbs, Lisa A. Grimes.

**Formal analysis:** Daniel J. Stubbs, Lisa A. Grimes.

**Methodology:** Daniel J. Stubbs.

**Project administration:** Daniel J. Stubbs, Lisa A. Grimes, Ari Ercole.

**Supervision:** Ari Ercole.

**Validation:** Daniel J. Stubbs, Ari Ercole.

**Writing – original draft:** Daniel J. Stubbs.

**Writing – review & editing:** Daniel J. Stubbs, Lisa A. Grimes, Ari Ercole.

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
