## [Decision Letter · Decision Letter 0]

2 Dec 2019

Performance of cardiopulmonary exercise testing for the prediction of post-operative complications in non cardiopulmonary surgery: a systematic review

PONE-D-19-24934

Dear Dr. Ercole,

We are pleased to inform you that your manuscript has been judged scientifically suitable for publication and will be formally accepted for publication once it complies with all outstanding technical requirements.

With kind regards,

Gabor Kovacs

Academic Editor

PLOS ONE

Journal Requirements:

1. During our internal evauation, the in-house editorial staff noticed that the date of the last electronic search was conducted on July 24th 2018. In order to ensure systematic reviews are comprehensive, unbiased and timely, the electronic database search should be conducted within the last 12 months of submission. At this time, we request that you please provide an updated search and include any new studies in your meta-analysis.

Additionally, please provide a description of any assessment of research/methodological quality of the studies included in your meta-analysis, such as the use of the Newcastle-Ottawa scale for non-randomised studies.

Additional Editor Comments (optional):

The editor and all reviewers appreciated the clinical importance of your manuscript. The incorporation of patients with a broad spectrum of surgical interventions belonged to one of the major strengths of the paper.  The editor and the reviewers believe that  the manuscript will be useful in recognizing patients not at increased peri-operative risk and will facilitate the process of shared decision making and the allocation of resources.

Reviewers' comments:

Reviewer's Responses to Questions

**Comments to the Author**

1. Is the manuscript technically sound, and do the data support the conclusions?

Reviewer #1: Yes

Reviewer #2: Yes

Reviewer #3: Yes

2. Has the statistical analysis been performed appropriately and rigorously? 

Reviewer #1: Yes

Reviewer #2: Yes

3. Have the authors made all data underlying the findings in their manuscript fully available?

Reviewer #1: Yes

Reviewer #2: Yes

Reviewer #3: Yes

4. Is the manuscript presented in an intelligible fashion and written in standard English?

Reviewer #1: Yes

Reviewer #2: Yes

Reviewer #3: Yes

5. Review Comments to the Author

Reviewer #1: Stubbs and colleagues performed a critical review about the clinical impact of cardiopulmonary exercise testing for the prediction of post-operative complications in non cardiopulmonary surgery. In my opinion this systematic review is well structured and stresses the various advantages and drawbacks concerning preoperative functional risk evaluation.

Reviewer #2: The authors made a huge effort to review the literature on the use of cpet for the prediction of post-operative

complications for cardiorespiratory complications, 30 day or in-hospital death, and unplanned intensive care unit admission for non-cardiopulmonary surgery from 1993 until now.

The review process including search strategy, inclusion and exclusion criteria, data extraction and prediction performance metrics is documented in detail, statistical analysis are robust.

The high negative predictive values (96-100%) and poor positive predictive value (2.9-42.1%) and differences as a predictor of mortality between anaerobic threshold, VE/VCO2 and peak VO2 might have implications on perioperative assessment startegies in real life and may be of interest in the integrative multidisciplinary approach of preoperative patients.

Therefore i would recommend to accept this paper.

6. PLOS authors have the option to publish the peer review history of their article (what does this mean?). If published, this will include your full peer review and any attached files.

Reviewer #1: No

Reviewer #2: No

Reviewer #3: No

---

## [Editor Report · Acceptance letter]

21 Jan 2020

PONE-D-19-24934

Performance of cardiopulmonary exercise testing for the prediction of post-operative complications in non cardiopulmonary surgery: a systematic review

Dear Dr. Ercole:

I am pleased to inform you that your manuscript has been deemed suitable for publication in PLOS ONE. Congratulations! Your manuscript is now with our production department.

With kind regards,

on behalf of

Dr. Gabor Kovacs

Academic Editor

PLOS ONE